# Learn to Select: Efficient Cross-device Federated Learning via Reinforcement Learning

**Chunlin Tian[1], Zhan Shi[2], Li Li[1]**
[1]University of Macau, IOTSC, [2]The University of Texas at Austin

## Abstract

Federated Learning (FL) is a collaborative training method that provides data privacy in the age of big data. However, it is often ineffective on edge devices due to their heterogeneous and constrained resources. The primary challenge is to identify devices with useful training data and available compute capability, which are both time-varying. In this paper, we propose FedRank, a novel federated learning approach based on reinforcement learning. Our approach addresses the device selection problem by casting it as a ranking problem and employing a pairwise training scheme. Furthermore, we leverage imitation learning from state-of-the-art algorithms to eliminate the cold start phenomenon that offsets the benefits of previous learning-based approaches. Experimental results show that FedRank improves model accuracy by 2.59%-12.75%, and accelerates the training process by up to $2.25\times$ and $1.76\times$ on average at the same time.

## 1 Introduction

**Motivation.** Federated learning (FL) is a powerful technique that allows resource-constrained mobile devices to collaborate on training a global machine learning model while preserving data privacy. Despite the growing number of applications, deploying FL across heterogeneous mobile devices in the real world poses several challenges. First, the heterogeneous training capabilities and runtime variations of the devices may lead to stragglers and significantly slow down the FL training process. Second, the heterogeneity of data across different devices may affect the convergence speed and performance of the global model.

**Limitations of the State of the Art.** Selecting the appropriate training devices is therefore critical for the success of FL. To this end, two lines of solutions have been proposed: heuristic-based and learning-based. Heuristic-based approaches focus on individual considerations in the process of device selection, such as heterogeneous data (Balakrishnan et al., 2022), training processes (Diao et al., 2021), and energy efficiency (Li et al., 2019). Recent work (Tian et al., 2022; Lai et al., 2021) have achieved state-of-the-art results by using analytical approaches that carefully addresses all aspects of device selection. However, heuristic-based approaches are not robust in scenarios they have not encountered before, and adapting to new and unseen scenarios often requires domain expertise and significant tuning. In contrast, learning-based approaches, such as AutoFL (Kim & Wu, 2021) and MARL (Zhang et al., 2022), can potentially select devices that balance all requirements in a data-driven fashion. However, they are sample-inefficient and suffer from the cold-start phenomenon due to the limited number of training rounds, resulting in lower quality options in early stages and limiting their usefulness in real-world scenarios.

**Key Insights.** This paper introduces FedRank, a novel approach for device selection in FL. Our solution is based on two key insights. First, we draw an analogy between device selection and Learning to Rank (LTR) problems, such as recommendation. Specifically, we aim to develop FedRank as a recommendation system that suggests the most valuable devices to the FL training agent. To achieve this, the recommendation system is trained to rank the values of all devices and selects the top-K. By adopting this view, we naturally replace the pointwise loss with pairwise loss, which has shown significant benefits. Second, we recognize that the cold-start phenomenon has hindered the practical application of RL-based solutions, leading to sub-optimal performance compared against analytical-based approaches. To address this problem, we propose a pretraining strategy for FedRank that utilizes a state-of-the-art analytical model via imitation learning. We then fine-tune FedRank using real-world interactions to surpass the performance of the analytical model.

## 2    FEDRANK

This section describes the proposed FedRank algorithm. Based on the insights discussed above, FedRank improves over SOTA through Learning to Rank (LTR) to improve the model training, and imitation learning to overcome the cold-start problem.

**The Ranking View of Device Selection.** In device selection, just like in recommendation, the absolute rewards of selecting devices are implicit and unimportant. Instead, the ranking of the rewards is critical since devices with the highest rewards will be selected for training. Therefore, our RL loss function adopts pairwise loss (Joachims, 2002) to emphasize the relative orders over the absolute values of rewards. For more detailed LTR design, please refer to Appendices C.

**Warmup with Imitation Learning.** In dynamic and heterogeneous training scenarios, traditional RL schemes suffer from a cold start problem (Schein et al., 2002), leading to poor performance the early training period (Zhang et al., 2022). To overcome this problem, we use imitation learning (Schaal, 1996) from SOTA heuristic-based approaches to improve the performance of previous learning-based approaches. For more detailed imitation learning, please refer to Appendices D.

| Dataset | FedAvg | FedMarl | Harmony | FedRank |
|---------|--------|---------|---------|---------|
| MNIST | 92.53% | 93.53% | 96.22% | 98.81% |
| CIFAR10 | 39.75% | 41.85% | 49.33% | 52.06% |

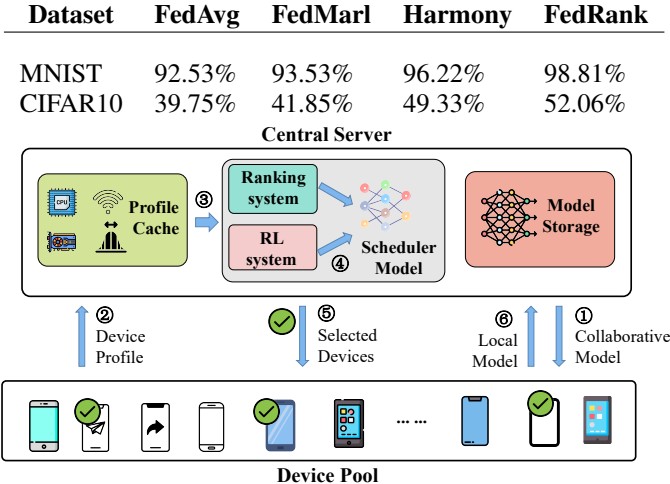

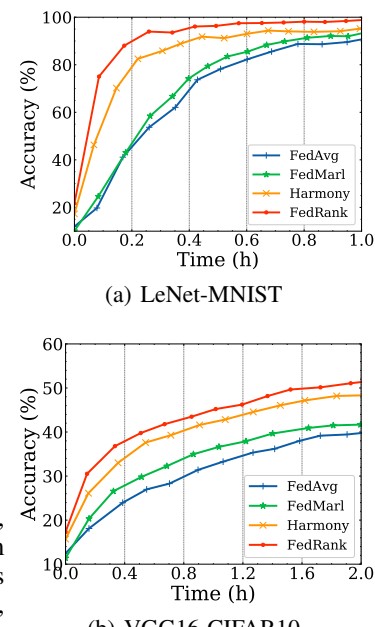

(a) LeNet-MNIST

(b) VGG16-CIFAR10

Figure 1: Workflow and architecture of FedRank. In each round, all mobile devices receive a copy of the collaborative model from **Model Storage** and perform the probing training. Then devices send the profile information to the **Profile Cache**. After that, the well-imitated learning **Scheduler Model** intelligently selects the *right* participating devices to perform follow-up training. See Appendices A, and B for more detailed steps and RL design, respectively.

Figure 2: Comparison of model accuracy-to-time.

## 3    EVALUATION

We evaluate the performance of FedRank with a hybrid testbed with both simulation and off-the-shelf mobile devices which effectively emulates the data and system heterogeneity in real-world cases. More details of the experimental setup are shown in Appendix E. Figure 2 depicts the evaluation results with LeNet5 (LeCun et al., 1998) on MNIST (Lecun et al., 1998) and VGG16 (Simonyan., 2014) on CIFAR10 (Krizhevsky et al., 2009) comparing against the following baselines: FedAvg (McMahan et al., 2017), FedMarl (SOTA learning-based) Zhang et al. (2022) and Harmony (SOTA heuristic-based) (Tian et al., 2022). FedRank achieves significantly faster convergence and higher accuracy. Specifically, FedRank achieves 98.81% and 52.06% test accuracy on MNIST and CIFAR10, which improves accuracy by 6.23% over FedAvg, 5.28% over FedMarl, 2.59% over Harmony on MNIST, and 12.75% over FedAvg, 10.21% over FedMarl, 2.73% over Harmony on CIFAR10. FedRank speeds up the training to convergence by 1.50× to 2.25× on MNIST, and 1.25× to 2.08× on CIFAR10.

URM STATEMENT

The authors acknowledge that at least one key author of this work meets the URM criteria of ICLR 2023 Tiny Papers Track.

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

## A  WORKFLOW OF FEDRANK.

The workflow for each round is as follows:

1. The mobile devices download a copy of the collaborative model from the **Model Storage**.
2. They perform the probing training (Zhang et al., 2022) (first epoch of local training) and send probing information (training latency and cost, communication bandwidth, loss) to the **Profile Cache**.
3. The Profile Cache fuses the information from each device and sends it to the **Scheduler Model**.
4. The Scheduler Model then acts as a heterogeneity-aware FL process scheduler that intelligently selects participating devices by jointly optimizing runtime trainability/performance and data homogeneity under user-specific constraints.
5. The coordination result is sent from the central server to the selected devices.
6. The selected devices complete local training and upload model updates to Model Storage, which then aggregates the local gradients to update the global model.

## B  REINFORCEMENT LEARNING DESIGN DETAILS

The following is an example of the reinforcement learning setup used in this paper, but FedRank has a high degree of flexibility and compatibility, allowing the user to tailor the setup to different exploration goals.

For RL, we use a DQN as the scheduler model. Specifically, in episode t, the DNN is used to implement the Q function $Q_i^\theta(s, a) = E[R_t | s_i^t = s, a_i^t = a]$, and the state $s_i^t$ obtained by profiling device $i$ acts as the input. After collecting the maximum Q-value (i.e. $argmax_a Q_i^\theta(s_i^t, a)$, where $\theta$ is the parameter of the DNN) of all devices, the top-K device is selected as specified by the user, and the action of the corresponding devices are set to $a_i^t = 1$, while the others are set to $a_i^t = 0$. When all devices have completed their DNN inference, the scheduler receives a common reward $r_t$ and proceeds to the next state $s_i^{t+1}$. A cache is used to store the tuple of transitions $< s_i^t, a_i^t, s_i^{t+1}, t_t >$ for each node $i$ in order to train the DNN. A shared Q-function $\boldsymbol{Q}(.)$ is represented as the element-wise sum of all separate Q-functions ( i.e, $\boldsymbol{Q}(\boldsymbol{s}_t, \boldsymbol{a}_t) = \sum_i Q_i^\theta(s_i^t, a_i^t)$, where $\boldsymbol{s}_t = s_i^t$ and $\boldsymbol{a}_t = a_i^t$ are aggregated from across all devices). Then, the DNN is trained recursively with minimum loss:

$$L = E_{\boldsymbol{s}^t, \boldsymbol{a}, r, \boldsymbol{s}^{t+1}}[r_t + \gamma * \sum_i Q_i^{\theta'}(\boldsymbol{s}_i^{t+1}, a) - Q_i^\theta(\boldsymbol{s}_i^t, \boldsymbol{a})] \tag{1}$$

where $\theta'$ denotes the target network parameters that were periodically copied from $\theta$ throughout the entire training phase.

### B.1  STATE

We employ probing training to profile the performance of local devices, and Zhang et al. (2022) demonstrates that probing training is effective in estimating and capturing the runtime state of devices at low cost. For device i, the state vector can be defined as:

$$\boldsymbol{s}_i^t = [T_{comp,i}^t, T_{com,i}^t, E_{comp,i}^t, E_{comm,i}^t, L_i^t, D_i^t] \tag{2}$$

where:

**System Heterogeneity:**

- Probe Training Latency $T_{comp,i}$;

- Communication Latency $T_{com,i}$;
- Computing Energy Cost $E_{comp,i}$;
- Communication Energy Cost $E_{comm,i}$;

**Data Heterogeneity:**

- Training Loss $L_i$;
- Size of Data $D_i$;

### B.2 ACTION

The cluster of selected participant devices is defined as the action in this RL model, which makes up a tunable knob of the system. It indicates the decision whether the device is to proceed training ($a_i^t = 1$) or be terminated prematurely ($a_i^t = 0$).

$$\boldsymbol{A}_N^t = \left\{a_1^t, a_2^t, \ldots, a_i^t\right\}, \quad a_i^t \in [0,1] \tag{3}$$

### B.3 REWARD

- **Energy** $E^t$. The total energy consumption of training round $t$ is calculated for a cluster of all $N$ participating devices as ( 4), where $l_e p$ denotes the # of local training epoch.

$$E^t = \sum_N (E_{comm,i}^t + E_{comp,i}^t) + \sum_N (E_{comm,i}^t + E_{comp,i}^t * (l\_ep - 1)) * a_n^t \tag{4}$$

- **Training latency** $T^t$. the processing latency of training round $t$ can be defined as ( 5).

$$T^t = \max_{i \in N}(T_{comm,i}^t + T_{comp,i}^t) + \max_{i \in N}(T_{comm,i}^t + T_{comp,i}^t * (l\_ep - 1)) * a_n^t \tag{5}$$

- **Accuracy** $Acc_t$. It denotes the test accuracy of global model at training each round.

The reward $r^t$ that jointly balance the model performance, training efficiency and energy-friendly at training round t is denotes as:

$$r^t = \omega_1 * \Delta ACC - \omega_2 * \max_{i \in K}(T_i) - \omega_3 * \sum_K E \tag{6}$$

## C  OPTIMIZATION OF THE RL LOSS FUNCTION BY LEARN-TO-RANK (PAIRWISE)

In the training process, for each device we are only interested in whether it participates in the current round, rather than its actual Q-value. While a pure RL scheme would over-emphasise irrelevant devices, we introduce LTR to optimize the loss function during DQN model training in to find the relevant devices through relevance. For this section, we adopt pairwise to rebulid the RL loss function in (2). We map from the DQN model outputs Q-value to probabilities using a logistic function as:

$$P_{i,j} = \sigma[Q(s_i, a_i) - Q(s_j, a_j)]$$
$$\overline{P_{i,j}} = \sigma[\bar{Q}(s_i, a_i) - \bar{Q}(s_j, a_j)] \tag{7}$$

After that, we use a binary cross entropy loss to represent the pairwise loss of device i and j as (8). And we can obtain the average pairwise loss $L_{Rank} = E[C_{i,j}]$.

$$C_{i,j} = -\overline{P_{i,j}} \log P_{i,j} - \left(1 - \overline{P_{i,j}}\right) \log (1 - P_{i,j}) \tag{8}$$

Therefore, the new joint loss function of RL DNN model can be defined as:

$$L = \overline{L_{RL}} + \epsilon * \overline{L_{Rank}} \tag{9}$$

While Figure 3 illustrates the detail of Scheduler Model workflow.

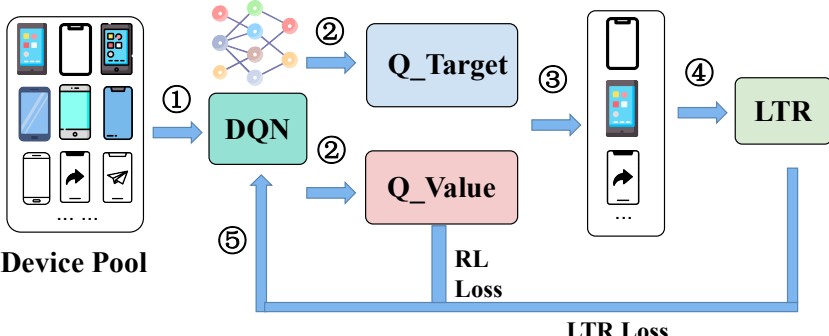

Figure 3: Workflow of Scheduler Model, steps are shown in circles.

## D    IMITATION LEARNING

Current RL suffers from the cold-start problem, where meaningfulness fails in the early stages of training. And heuristic methods that specify specific utility functions fail in diverse and heterogeneous real-world deployment environments. Therefore, we propose imitation learning to solve the cold-start problem in the early stages of reinforcement learning. Even though RL cannot be applied directly, we use well-known heuristics as conditions to warm up RL, making it more accurate in selecting participant devices that match the user-specific ones even in diverse and complicated situations.

First, we convert a given sequence of mobile devices into states $\{s_0, ... , s_t\}$, on which we can compute the utility values and select the optimal participating devices with Harmony (Tian et al., 2022). Then, with the given states, we train FedRankith the heterogeneous context and compute the loss value $L$, which encourages the learned selection policy to make the same decisions as Harmony. Finally, we apply the warm-up RL model to the online optimization process.

## E    EXPERIMENTAL METHODOLOGY

Specifically, we first build a simulator following the server/client architecture based on TensorFlow Federated (TFF) (Abadi et al., 2022), in which different processes are created to emulate the central server and the participating devices with heterogeneous data distribution. In addition, we implement the local training process with MNN (Jiang et al., 2020) on four types of mobile phones with diverse hardware configurations, including Google Pixel 6 (Google Tensor), OnePlus 10 pro (Snapdragon 8 Gen 1), Redmi Note 10 (Snapdragon 678) and Realme Q3s (Snapdragon 778G). The profiled local training completion time in the corresponding cases (different amount of training data, various concurrently running apps) are integrated into the simulator to emulate the system heterogeneity.

Common settings:

- # of devices in pool N: 100;
- # of local training epoch: 5;
- # of local training batch size: 64;
- # Non-IID parameter of Dirichlet distribution: 0.5;
- Learning rate of TFF: client_lr=0.02, and server_lr=1.0;

Other settings:

- FedAvg: randomly select 10 participating devices at each round.
- FedMarl (the SOTA learning-based FL system with multi-agent reinforcement learning): randomly select 20 devices for the agents to make further optimization decisions about participating devices. We make the hyperparameters of the reward function as: $\omega_1 = 1.0$, $\omega_2 = 0.2$ and $\omega_3 = 0.1$.

- Harmony (the SOTA heuristic-based FL system.) we select the optimal top-10 participating devices based on the utility value
- FedRank : we select the optimal top-10 participating devices. We set the developer-specified parameters as: $\epsilon = 0.1$. And we use Harmony to offline train RL with 200 episodes.

