# OpenReview forum: "Learn to Select: Efficient Cross-device Federated Learning via Reinforcement Learning"
_ICLR.cc/2023/TinyPapers — Submitted to Tiny Papers @ ICLR 2023_

### Official Review · Reviewer_m7fr · 2023-03-24

**Confidence:** 4

**Summary Of Contributions:**

The paper proposes a device selection algorithm in the federated learning setup. For that, the authors have proposed two fold strategy: one is based on device selection and second is their ranking.

**Rating:**

Great Start (GS): a submission which meets some of the reviewing criteria but has room for improvement

**Strengths And Weaknesses:**

The selection of devices in the learning of the global federated model is an important direction. This paper proposes a novel algorithm for the selection of clients based on the reward function in the RL setting.

It is not clear from the following text: "However, heuristic-based approaches are not robust in scenarios they have not encountered before, and adapting to new and unseen scenarios often requires domain expertise and significant tuning" what kind of scenarios authors are referring and what and why are the limitations?

The ranking algorithm is not clear. How the rewards has been defined? What if the selected client is not available due to computational power or any other issue? How many clients should be selected at least to learn the models? In each iteration are the top-K clients are same or different? Are the selected clients solve the task of all the clients involved?

What is the extra computational cost involved in the ranking of clients and running the RL algorithm?

**Suggested Changes:**

The reproducibility of the paper is limited due to the lack of several important pieces of information as mentioned in the weakness section.
The authors also need to justify the use of RL and device selection especially when the FL learning itself is time-consuming. What is the convergence rate of the proposed algorithm?
Comparison with similar algorithms mentioned including heuristic-based ones in the literature is missing.
How the rewards functions are designed and decided?
Few typos: (SOTA earning-based)

---

### Official Review · Reviewer_yCqo · 2023-04-02

**Confidence:** 4

**Summary Of Contributions:**

This paper proposes a new method (FedRank) based on reinforcement learning to automate the device selection process in federated learning. RL methdology in this paper has been improved by warmup in imitation learning. The device selection problem has been reduced to a Learning-to-Rank problem.

**Rating:**

High Potential (HP): a submission which meets the reviewing criteria and has potential to make an impact on the field

**Strengths And Weaknesses:**

**Strengths:**

S1: The idea of using RL to optimize the device selection process in federated learning is novel. As far as I know, this is the first work. The formalization of the problem is also clear.

S2: This paper is well written. The logic is easy to follow. All the findings in the paper is expressed clearly and effectively.

S3: Experiments including both simulation and real-world mobile devices are strong and the experimental results are good enough to cover the statements in the paper. The comparison to multiply FL baselines is a plus.

S4:The claims and conclusions are justified by the findings. Authors follow the formatting requirements.

**Weaknesses:**

W1: The organization of the paper is not good. Authors should leave space for conclusion and squeeze the space for introduction.

W2: Could the authors talk more about the setup of the experiments? e.g. how many devices/how much percentage of the devices are selected in a communication round?

W3: Could authors release their source code?

**Suggested Changes:**

C1: The organization of this paper can be improved by adding the conclusion and reduce the size of introduction.

C2: Please cite more papers related to this field:

[1] Optimizing federated learning on non-iid data with reinforcement learning

[2] Federated Multiple Label Hashing (FedMLH): Communication Efficient Federated Learning on Extreme Classification Tasks

[3] Client selection in federated learning: Convergence analysis and power-of-choice selection strategies

---

### Meta-Review · Area_Chair_D6Pn · 2023-04-08

**Recommendation:** Invite to archive
**Confidence:** 4

**Metareview:**

The paper proposes a novel method for automating the device selection process in federated learning using reinforcement learning.

Quality:
The paper is of good quality, with strong experiments covering both simulation and real-world mobile devices. However, there are concerns raised about the lack of information on the ranking algorithm and reproducibility.

Clarity:
The paper is well written and easy to follow, with clear statements and justified conclusions. However, there are suggestions to improve the organization of the paper by adding a conclusion and reducing the size of the introduction.

Originality:
The use of reinforcement learning to optimize the device selection process in federated learning is considered novel. However, there are questions raised about the justification for using RL and device selection, and suggestions to compare with similar algorithms in the literature.

Significance:
The paper has the potential to make an impact on the field, with novel contributions to the device selection process in federated learning. However, there are concerns raised about the lack of information on the ranking algorithm and reproducibility, and questions on the justification for using RL and device selection, and comparison with similar algorithms in the literature.


**Summary:**

a novel method for automating the device selection process in federated learning using reinforcement learning

**Comments And Feedback To The Authors:**

Both reviewers have provided constructive feedback and suggestions for improvement, and it would be helpful for the authors to address these concerns in a revised version of the paper.

**Reason For Not Giving A Higher Recommendation:**

There are some concerns raised about the paper, including the lack of information on the ranking algorithm and reproducibility, and questions on the justification for using RL and device selection, and comparison with similar algorithms in the literature.

**Reason For Not Giving A Lower Recommendation:**

The paper has several strengths, including its novel use of reinforcement learning for device selection in federated learning, strong experiments covering both simulation and real-world mobile devices, clear and effective findings, and justified claims and conclusions. Although there are some concerns raised about the lack of information on the ranking algorithm and reproducibility, and questions on the justification for using RL and device selection, and comparison with similar algorithms in the literature, these can be addressed in a revised version of the paper. Overall, the paper has the potential to make an impact on the field and is considered a good start with room for improvement.

---

### Decision · Program_Chairs · 2023-04-09

Invite to archive